1. **Localising individual atoms of tryptophan side chains in the metallo-β-lactamase IMP-1**
2. **by pseudocontact shifts from paramagnetic lanthanoid tags at multiple sites**

3. Henry W. Orton,[a,*] Iresha D. Herath,[b,*] Ansis Maleckis,[c] Shereen Jabar,[b] Monika Szabo,[d] Bim
4. Graham,[d] Colum Breen,[e] Lydia Topping,[e] Stephen J. Butler,[e] Gottfried Otting[a]

5.

6. [a] ARC Centre of Excellence for Innovations in Peptide & Protein Science, Research School of
7. Chemistry, Australian National University, Canberra, ACT 2601, Australia

8. [b] Research School of Chemistry, The Australian National University, Sullivans Creek Road,
9. Canberra ACT 2601, Australia

10. [c] Latvian Institute of Organic Synthesis, Aizkraukles 21, LV-1006 Riga, Latvia

11. [d] Monash Institute of Pharmaceutical Sciences, Monash University, Parkville, VIC 3052,
12. Australia

13. [e] Department of Chemistry, Loughborough University, Epinal Way, Loughborough, LE11
14. 3TU, United Kingdom

18.

19. Correspondence: Gottfried Otting (gottfried.otting@anu.edu.au)

20. [*] These authors contributed equally to this work.

21.

22. **Abstract**

23. The metallo-β-lactamase IMP-1 features a flexible loop near the active site that assumes
24. different conformations in single crystal structures, which may assist in substrate binding and
25. enzymatic activity. To probe the position of this loop, we labelled the tryptophan residues of
26. IMP-1 with 7-$^{13}$C-indole and the protein with lanthanoid tags at three different sites. The
27. magnetic susceptibility anisotropy ($\Delta\chi$) tensors were determined by measuring pseudocontact
28. shifts (PCS) of backbone amide protons. The $\Delta\chi$ tensors were subsequently used to identify
29. the atomic coordinates of the tryptophan side chains in the protein. The PCSs were sufficient
30. to determine the location of Trp28, which is located in the active site loop targeted by our
31. experiments, with high accuracy. Its average atomic coordinates showed barely significant
32. changes in response to the inhibitor captopril. It was found that localisation spaces could be
33. defined with better accuracy by including only the PCSs of a single paramagnetic lanthanoid
34. ion for each tag and tagging site. The effect was attributed to the shallow angle with which

PCS isosurfaces tend to intersect if generated by tags and tagging sites that are identical except
for the paramagnetic lanthanoid ion.

**1 Introduction**
The metallo-β-lactamase IMP-1 is an enzyme that hydrolyses β-lactams, thus conferring
penicillin resistance to bacteria. First identified 30 years ago in the Gram-negative bacteria in
early 1990s from *Pseudomonas aeruginosa* and *Serratia marcescens* (Bush 2013)*,* IMP-1 has
become a serious clinical problem due to horizontal gene transfer by a highly mobile gene
($bla_{IMP-1}$) located on an integron (Arakawa et al., 1995), as the $bla_{IMP-1}$ gene has been detected
in isolates of *Klebsiella pneumoniae*, *Pseudomonas putida*, *Alcaligenes xylosoxidans*,
*Acinetobacter junii*, *Providencia rettgeri*, *Acinetobacter baumannii* and *Enterobacter*
*aerogenes* (Ito et al., 1995; Laraki et al., 1999a; Watanabe et al., 1991). Critically, IMP-1
confers resistance also to recent generations of carbapenems and extended-spectrum
cephalosporins (Laraki et al., 199b; Bush et al., 2010; van Duin et al., 2013).

Multiple crystal structures have been solved of IMP-1, free and in complex with various

inhibitors (Concha et al., 2000; Toney et al., 2001; Moali et al., 2003; Hiraiwa et al., 2014;
Brem et al., 2016; Hinchliffe et al., 2016; 2018; Wachino et al., 2019; Rossi et al., 2021). IMP-
1 belongs to the subclass B1 of metallo-β-lactamases, which contain two zinc ions bridged by
the sulfur atom of a cysteine residue in the active site (Concha, 2000). One of $Zn^{2+}$ ions can
readily be replaced by a $Fe^{3+}$ ion (Carruthers et al., 2014). The active site is flanked by a loop
(referred to as L3 loop) that contains a highly solvent-exposed tryptophan residue surrounded
by glycine residues on either side. Both the loop and the tryptophan residue (Trp28 in the IMP-
1-specific numbering used by Concha et al. (2000) and Trp64 in the universal numbering
scheme by Galleni et al. (2001)) assume different conformations in different crystal structures,
suggesting that the loop acts as a mobile flap to cover bound substrate (Fig. 1A). The L3 loop
and the functional implication of its flexibility has been studied extensively for different
metallo-β-lactamases containing the Gly-Trp-Gly motif in the loop (Huntley et al., 2000; 2003;
Moali et al., 2003; Yamaguchi et al., 2015; Palacios et al., 2019; Gianquinto et al., 2020; Softley
et al., 2020).  Flexibility of the L3 loop is a general feature also of many metallo-β-lactamases
without the Gly-Trp-Gly motif and is thought to contribute to the wide range of β-lactam
substrates that can be hydrolyzed by the enzymes (González et al., 2016; Linciano et al., 2019;
Salimraj et al., 2018). In the case of the metallo-β-lactamase from *B. fragilis*, which is closely
related to IMP-1, electron density could be detected for the Gly-Trp-Gly motif in the crystal
structure of the protein in the presence (Payne et al., 2003) but not absence of an inhibitor
(Concha et al., 1996), and an NMR relaxation study in solution confirmed the increased
flexibility of both the L3 loop and, in particular, the sidechain of the tryptophan residue
(Huntley et al., 2000). A similar situation prevails in the case of the IMP-1 variant IMP-13,
where different crystal structures of the ligand-free protein show the L3 loop in very different
conformations, sometimes lacking electron density, while NMR relaxation measurements
confirmed the increased flexibility of the loop (Softley et al., 2020).
Due to the rigidity of their sidechains, tryptophan residues frequently contribute to the
structural stability of three-dimensional protein folds and it is unusual to observe tryptophan
sidechains fully solvent-exposed as in the Gly-Trp-Gly motif of substrate-free IMP-1. The
functional role of Trp28 in IMP-1 was assessed in an early mutation study by mutating Trp28
to alanine and, in a different experiment, eliminating the L3 loop altogether. Enzymatic activity
measurements revealed an increase in the Michaelis constant $K_m$ and a decrease in $k_{cat}/K_m$ ratios
for all β-lactams tested, illustrating the importance of the Trp28 sidechain for catalytic activity.
Complete removal of the L3 loop reduced the $k_{cat}/K_m$ ratios even further, but without
completely abolishing the enzymatic activity (Moali et al., 2003).


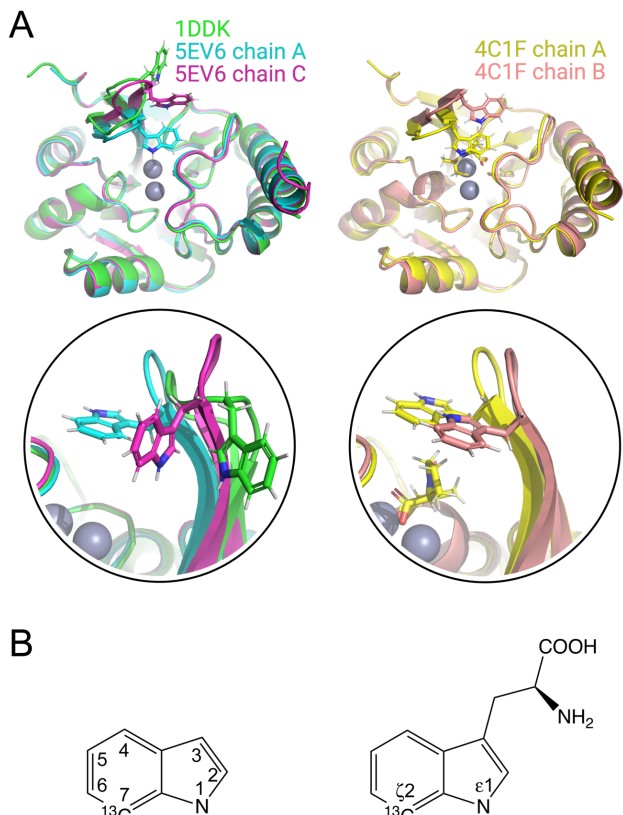


**Figure 1.** Crystal structures of IMP-1 with different conformations of the loop L3 and chemical structures of indole and tryptophan with atom names. (A) Superimposition of crystal structures of IMP-1 highlighting structural variations of Trp28 and the associated loop L3. The structures shown are of the $Zn^{2+}/Zn^{2+}$ complex without inhibitor (green, PDB ID 1DDK, Concha et al., 2000; cyan for chain A and magenta for chain C, PDB ID 5EV6, Hinchliffe et al., 2016), with bound L-captopril (yellow for chain A and salmon for chain B, PDB ID 4CIF, Brem et al., 2016). $Zn^{2+}$ ions are represented by grey spheres and bound captopril is shown in the structure 4C1F chain A. (B) Chemical structures of indole and tryptophan with selected ring positions labelled according to IUPAC conventions. The present work used indole synthesised with a $^{13}C$-$^{1}H$ group in position 7 and deuterium in the ring positions 2, 3, 4, 5 and 6 (Maleckis et al., 2021).

In the crystalline state, the conformation of a solvent-exposed loop is easily impacted by crystal packing forces. Therefore, it is unclear what the actual conformation of the L3 loop is in solution. To address this question, we used solution NMR spectroscopy to assess the location of Trp28 in IMP-1 both in the absence and presence of the inhibitor L-captopril, which inhibits metallo-β-lactamases by binding to the active-site zinc ions (Brem et al., 2016). The analysis was hindered by incomplete backbone resonance assignments of IMP-1 attributed to conformational exchange processes in parts of the protein (Carruthers et al., 2014). As it is difficult to accurately position the atoms of a solvent-exposed polypeptide loop in solution by nuclear Overhauser effects (NOE), we used pseudocontact shifts (PCS) generated by lanthanoid ions attached at different sites of IMP-1 to determine the location of Trp28 relative to the core of the protein. PCSs generated by multiple different paramagnetic metal ions or the same metal ion attached at different sites of a protein have previously been shown to allow localising atoms at remote sites of interest, such as in specific amino acid side chains (Pearce et al., 2017; Lescanne et al., 2018), bound ligand molecules (Guan et al., 2013; Chen et al., 2016) or proteins (Pintacuda et al., 2006; Keizers et al., 2010; de la Cruz et al., 2011; Kobashigawa et al., 2012; Brewer et al., 2015) or for 3D structure determinations of proteins (Yagi et al., 2013; Crick et al., 2015; Pilla et al., 2017).

IMP-1 contains six tryptophan residues, each containing several aromatic hydrogens with similar chemical shifts. To increase the spectral resolution in the 2D NMR spectra recorded for PCS measurements, we labelled each tryptophan sidechain with a single $^{13}C$ atom by expressing the protein in the presence of 7-$^{13}C$-indole (Fig. 1B; Maleckis et al., 2021). The

results show that the localisation spaces defined by the tryptophan PCSs fully agree with
previously determined crystal structures of IMP-1 for all tryptophan residues. They suggest
little change in the average conformation of the L3 loop upon binding of captopril. The results
illustrate the accuracy with which the positions of individual atoms can be determined by PCSs
from lanthanoid tags even in proteins of limited stability.
**2 Experimental procedures**
**2.1 Production, purification and tagging of proteins**
**2.1.1 Plasmid constructs and $^{13}$C-labelled indole**
Three different cysteine mutations (A53C, N172C and S204C) were introduced into the *bla*$_{IMP1}$
gene in the pET-47b(+) plasmid using a modified QuikChange protocol (Qi and Otting, 2019).
Deuterated 7-$^{13}$C-indole was synthesized as described with deuteration in all positions other
than position 7 (Maleckis et al., 2021). The amino acid sequence of the protein was that
reported in the crystal structure 4UAM (Carruthers et al., 2014), except that the N-terminal
alanine residue was substituted by a methionine to avoid heterogeneity by incomplete
processing by amino peptidase.
**2.1.2 Protein production**
Uniformly $^{15}$N-labelled samples of the cysteine mutants of IMP-1 were expressed in *E. coli*
BL21(DE3) cells. The cells were grown at 37 °C in Luria–Bertani (LB) medium containing 50
mgL$^{-1}$ kanamycin until the OD$_{600}$ reached 0.6–0.8 and were then transferred to 300 mL of M9
medium (6 gL$^{-1}$ Na$_2$HPO$_4$, 3 gL$^{-1}$ KH$_2$PO$_4$, 0.5 gL$^{-1}$ NaCl, pH 7.2) supplemented with 1 gL$^{-1}$
of $^{15}$NH$_4$Cl. After induction with isopropyl-β-D-thiogalactopyranoside (IPTG, final
concentration 1 mM), the cells were incubated at room temperature for 16 hours. Following
centrifugation, the cells were resuspended in buffer A (50 mM HEPES, pH 7.5, 100 μM ZnSO$_4$)
for lysis by a homogeniser (Avestin Emulsiflex C5). The supernatant of the centrifuged cell
lysate was loaded onto a 5 mL SP column, the column was washed with 20 column volumes
buffer B (same as buffer A but with 50 mM NaCl) and the protein was eluted with a gradient
of buffer C (same as buffer A but with 1 M NaCl).
IMP-1 samples containing 7-$^{13}$C-tryptophan were produced by continuous exchange
cell-free protein synthesis (CFPS) from PCR-amplified DNA with eight-nucleotide single-
stranded overhangs as described (Wu et al., 2007), using 7-$^{13}$C-indole as a precursor for the *in*
*vitro* production of tryptophan (Maleckis et al., 2021). The CFPS reactions were conducted at
30 °C for 16 h using 1 mL inner reaction mixture and 10 mL outer buffer. Tryptophan was
omitted from the mixture of amino acids provided and deuterated 7-$^{13}$C-indole was added from
a stock solution in 50 % DMSO/50 % $H_2O$ to the inner and outer buffers at a final concentration
of 0.75 mM. The protein samples were purified as described above. About 5 mg of the indole
was required for preparing each NMR sample.
**2.1.3 Ligation with C2-Ln$^{3+}$ tag**
To ensure the reduced state of cysteine thiol groups, the protein samples were treated with 2
mM dithiothreitol (DTT) for 1 hour. Subsequently, the DTT was removed using an Amicon
ultrafiltration centrifugal tube with a molecular weight cut-off of 10 kDa, concentrating the
protein samples to 50 μM in buffer A. The samples were incubated overnight at room
temperature with shaking in the presence of five-fold molar excess of C2 tag (Graham et al.,
2011; de la Cruz et al., 2011) loaded with either Y$^{3+}$, Tb$^{3+}$ or Tm$^{3+}$. Following the tagging
reaction, the samples were washed using an Amicon centrifugal filter unit to remove unbound
tag and the buffer was exchanged to NMR buffer (20 mM MES, pH 6.5, 100 mM NaCl).
**2.1.4 Ligation with C12-Ln$^{3+}$ tag**
The ligation reaction of IMP-1 N172C with the C12-Ln$^{3+}$ tag loaded with either Y$^{3+}$, Tb$^{3+}$ or
Tm$^{3+}$ (Herath et al., 2021) was conducted in the same way as with the C2-Ln$^{3+}$ tags, except that
the reactions were carried out in buffer A with the pH adjusted to 7.0.
**2.2 NMR spectroscopy**
All NMR data were acquired at 37 °C on Bruker 600 and 800 MHz NMR spectrometers
equipped with TCI cryoprobes designed for 5 mm NMR tubes, but only 3 mm NMR tubes were
used in this project. Protein concentrations were 0.6 mM and 0.2 mM for $^{15}$N-HSQC spectra
of samples labelled with the C2 and C12 tag, respectively. The protein concentrations were 0.4
mM for $^{13}$C-HSQC and NOE-relayed $^{13}$C-HSQC spectra. $^{15}$N-HSQC spectra were recorded at
a $^1$H-NMR frequency of 800 MHz with $t_{1max}$ = 40 ms, $t_{2max}$ = 170 ms, using a total recording
time of 3 h per spectrum. $^{13}$C-HSQC spectra were recorded with a S$^3$E filter to select the low-
field doublet component due to the $^1J_{HC}$ coupling of the $^{13}$C-labelled tryptophan side chains.
The pulse sequence is shown in Fig. S9 and the spectra were recorded at a $^1$H-NMR frequency
of 600 MHz using $t_{1max}$ = 20–50 ms, $t_{2max}$ = 106 ms and total recording times of 2 h per
spectrum. $^{13}$C-HSQC spectra with NOE relay were recorded without decoupling in the $^{13}$C-
dimension, relying on relaxation and $^{13}$C equilibrium magnetisation to emphasize the narrow
doublet component. The NOE mixing time was 150 ms and the total recording time 3 h per
spectrum. The pulse sequence is shown in Fig. S10.
To account for uncertainties in concentration measurements, samples with L-captopril
were prepared with a nominal ratio of captopril to protein of 1.5:1. In the case of samples
tagged with the C2 tag, however, this lead to gradual release of some of the tag, as captopril
contains a free thiol group and the disulfide linkage of the C2 tag is sensitive to chemical
reduction. To limit this mode of sample degradation, the NOE-relayed $[^{13}C,^1H]$-HSQC spectra
were recorded with a smaller excess of captopril.

**2.3 $\Delta\chi$-tensor fits**
The experimental PCSs ($\Delta\delta^{PCS}$) were measured in ppm as the amide proton chemical shift
observed in NMR spectra recorded for the IMP-1 mutants A53C, N172C and S204C tagged
with $Tm^{3+}$ or $Tb^{3+}$ tags minus the corresponding chemical shift measured of samples made with
$Y^{3+}$ tags. The resonance assignments of the wild-type $Zn_2$ enzyme (BMRB entry 25063) were
used to assign the $^{15}N$-HSQC cross-peaks in the diamagnetic state. The program Paramagpy
(Orton et al., 2020) was used to fit magnetic susceptibility anisotropy ($\Delta\chi$) tensors to crystal
structures of IMP-1 solved in the absence and presence of the inhibitor captopril.

**3 Results**
**3.1 Protein production**
Three cysteine mutants of uniformly $^{15}N$-labelled IMP-1 were produced *in vivo*, where cysteine
residues replaced Ala53, Asn172 and Ser204, respectively. The purified proteins were tagged
with C2 tags containing $Tb^{3+}$ or $Tm^{3+}$ as the paramagnetic ions and $Y^{3+}$ as the diamagnetic
reference. Samples of the uniformly $^{15}N$-labelled mutant N172C were also ligated with C12
tags containing the same set of metal ions. The chemical structures of the tags are depicted in
Fig. S1. To record $^{13}C$-$^1H$ correlation spectra of the tryptophan side chains with minimal
spectral overlap, additional samples of the cysteine mutants were produced with selectively
$^{13}C$-labelled tryptophan residues. These samples were produced by cell-free protein synthesis
in the presence of 7-$^{13}C$ indole, deuterated except at the 7 position, with the omission of
tryptophan, using a recently established protocol (Maleckis et al., 2021). The residual activity
of tryptophan synthase in the cell-free extract was sufficient to produce tryptophan from the
added $^{13}C$-labelled indole. The resulting tryptophan residues contained a $^{13}C$-$^1H$ group in
position 7 ($^{13}C^{\zeta 2}$ and $^1H^{\zeta 2}$ in IUPAC nomenclature; Markley et al., 1998) and deuterons at all
other hydrogen positions of the indole ring except for the $H^N$ atom ($H^{\varepsilon 1}$ in IUPAC
nomenclature). The cell-free expression yielded about 2 mg of purified protein per millilitre of
inner cell-free reaction mixture. Mass spectrometry indicated that the tryptophan residues of
IMP-1 were $^{13}C/^2H$-labelled with about 80 % labelling efficiency at each of the six tryptophan
positions (Fig. S2). The purified proteins were ligated with C2-$Ln^{3+}$ tags containing either $Tb^{3+}$,
$Tm^{3+}$ or $Y^{3+}$ as in the case of the $^{15}N$-labelled samples. Ligation yields with the C2 tags were
practically complete as indicated by mass spectrometry (Fig. S2). The ligation yield of the
N172C mutant with C12 tags was about 90 % (Herath et al., 2021).

**3.2 NMR experiments and resonance assignments**
[$^{15}N,^1H$]-HSQC spectra were measured of the tagged proteins in the free state and in the
presence of L-captopril (Fig. S3–S8). $^1H$ PCSs of backbone amide protons measured in these
spectra were used to establish the $\Delta\chi$ tensors relative to the protein. The resonance assignment
of the [$^{15}N,^1H$]-HSQC spectra in the presence of inhibitor was transferred from the
corresponding spectra recorded in the absence of inhibitor. As no resonance assignments could
reliably be made in this way in areas of spectral overlap, fewer resonance assignments were
available in the presence than absence of inhibitor. Furthermore, due to captopril releasing
some of the C2 tags from the protein by breaking the disulfide bridge of the tag attachment,
spectra recorded in the presence of captopril contained additional cross-peaks from
diamagnetic protein.
To obtain tagged protein that is inert against chemical reduction, we also attached the
C12 tag to the mutant N172C. This tag, however, caused the appearance of additional peaks in
the [$^{15}N,^1H$]-HSQC spectra (Fig. S7). The additional peaks appeared in different sample
preparations, indicating sample degradation or perturbation of the local protein structure by the
tag. We therefore based the rest of the work mainly on the PCSs obtained with the C2 tags.
Tables S1 and S2 list the PCSs of the backbone amides measured in the absence and presence
of captopril.
$^1H$ PCSs of the tryptophan $H^{\zeta 2}$ protons were measured in [$^{13}C,^1H$]-HSQC spectra
recorded with $S^3E$ spin-state selection element (Meissner et al., 1997) in the $^{13}C$ dimension to
select the slowly relaxing components of the doublets split by $^1J_{HC}$ couplings. Cross-peaks were
observed for all six tryptophan residues except for the mutant N172C, which displayed cross-
peaks of only five tryptophan indoles (Fig. 2). The missing signal was attributed to Trp176
because of its close proximity to the tagging site. The indole $H^{\varepsilon 1}$ proton is located within 2.9 Å
of the H$^{\zeta 2}$ proton and the NOE between both protons was readily observed in a [$^{13}$C,$^1$H]-HSQC
experiment with NOE relay (Fig. 2). The H$^{\varepsilon 1}$ chemical shifts afforded better spectral resolution
than the H$^{\zeta 2}$ resonances. Comparison of the predicted and observed PCSs yielded resonance
assignments of all tryptophan H$^{\varepsilon 1}$ cross-peaks with particular clarity in the NOE-relayed
[$^{13}$C,$^1$H]-HSQC spectrum (Fig. 2). In addition, the assignment was supported by paramagnetic
relaxation enhancements (for example, Trp88 is near residue 53 and therefore its cross-peaks
were strongly attenuated in the paramagnetic samples of the A53C mutant). Different PCSs
were observed for all six tryptophan sidechains and different PCSs were observed for the H$^{\zeta 2}$
and H$^{\varepsilon 1}$ protons within the same indole sidechain. Each of the tryptophan sidechains showed
PCSs in most, if not all, of the mutants. As the L3 loop is near residue 172, the mutant N172C
endowed Trp28 with particularly large PCSs. Tables S3 and S4 report the PCSs measured in
this way for the samples labelled with C2 tags.

In contrast, assigning the indole N-H groups in the [$^{15}$N,$^1$H]-HSQC spectra was much

more difficult because IMP-1 is a protein prone to showing more than a single peak per proton
(Figs S5 and S6). In particular, the [$^{15}$N,$^1$H]-HSQC spectrum of wild-type IMP-1 selectively
labelled with $^{15}$N-tryptophan displayed six intense and at least three weak N$^{\varepsilon 1}$–H$^{\varepsilon 1}$ cross-peaks
(Fig. S6; Carruthers et al., 2014) and the [$^{15}$N,$^1$H]-HSQC spectra of the tagged cysteine mutants
showed evidence of heterogeneity too (Fig. S5). Nonetheless, the six most intense N$^{\varepsilon 1}$–H$^{\varepsilon 1}$
cross-peaks could be assigned by comparison to the PCSs observed in the NOE-relayed
[$^{13}$C,$^1$H]-HSQC spectrum and this assignment was used to measure the PCSs of the tryptophan
H$^{\varepsilon 1}$ resonances in the mutant N172C tagged with C12 tag (Fig. S8; Table S4).

Spectra recorded in the presence of L-captopril were very similar to those recorded

without the inhibitor, except that some new, narrow C-H cross-peaks appeared in the [$^{13}$C,$^1$H]-
HSQC spectra of the mutants A53C and S204C, which were suggestive of protein degradation
(Fig. 3). We consequently used the better-resolved indole H$^N$ cross-peaks to identify the correct
parent C-H cross-peaks. The chemical shifts of the tryptophan sidechains changed very little
in response to the presence of L-captopril, except for the $^{13}$C-chemical shift of Trp28, which is
nearest to the ligand binding site. The PCSs of the indole protons measured in the presence of
the inhibitor are listed in Tables S5 and S6.

**3.2 Δχ-tensor fits**
The Δχ-tensor parameters were determined using the program Paramagpy (Orton et al., 2020),
using all available $^1$H PCSs measured of backbone amides. Comparing the Δχ tensor fits to the
crystal structures 5EV6 chains A and C (Hinchliffe et al., 2016) and 1DDK (Concha et al.,
2000) of the free protein, the chain A of the structure 5EV6 proved to produce the smallest $Q$
factor by a small margin (Fig. S11) and was used as the reference structure of the free protein
for the subsequent evaluation. Similarly, chain A of the co-crystal structure published with the
inhibitor L-captopril (PDB ID: 4C1F; Brem et al., 2016) on average delivered better fits than
chain B and was used as the reference structure for the NMR data recorded in the presence of
L-captopril. The $\Delta\chi$-tensor fits of each mutant and tag used a common metal position for the
data obtained with the $Tb^{3+}$ and $Tm^{3+}$ tags. The fits positioned the paramagnetic centres at
distances between 8.2 and 9.4 Å from the $C^\beta$ atom of the tagged cysteine residues, which is
compatible with the chemical structure of the C2-tag. Figure 4 shows the correlations between
back-calculated and experimental PCSs and Table S7 reports the fitted $\Delta\chi$ tensor parameters.
Very similar $Q$ factors were obtained when using the PCSs measured in the absence of inhibitor
to fit the $\Delta\chi$ tensor to the co-crystal structure 4C1F or the PCSs measured in the presence of
inhibitor to fit the $\Delta\chi$ tensor to the crystal structure of the free protein. This indicates that the
protein structure did not change very much in response to inhibitor binding. This conclusion
was also indicated by the similarity between the backbone PCSs observed with and without
inhibitor (Fig. S12).

The $\Delta\chi$ tensors obtained with the $Tb^{3+}$ tags were larger than those obtained with the

$Tm^{3+}$ tags, which is also reflected by the consistently larger PCSs observed in the $^{13}C$-$^1H$
correlation spectra of Fig. 2 and 3. The fits of $\Delta\chi$ tensors to the protein backbone also yielded
better $Q$ factors for PCSs generated by $Tb^{3+}$ than $Tm^{3+}$ ions. Therefore, we determined the
localisation spaces of the tryptophan sidechains in the first instance by using their $^1H$ PCSs
measured with $Tb^{3+}$ tags only.

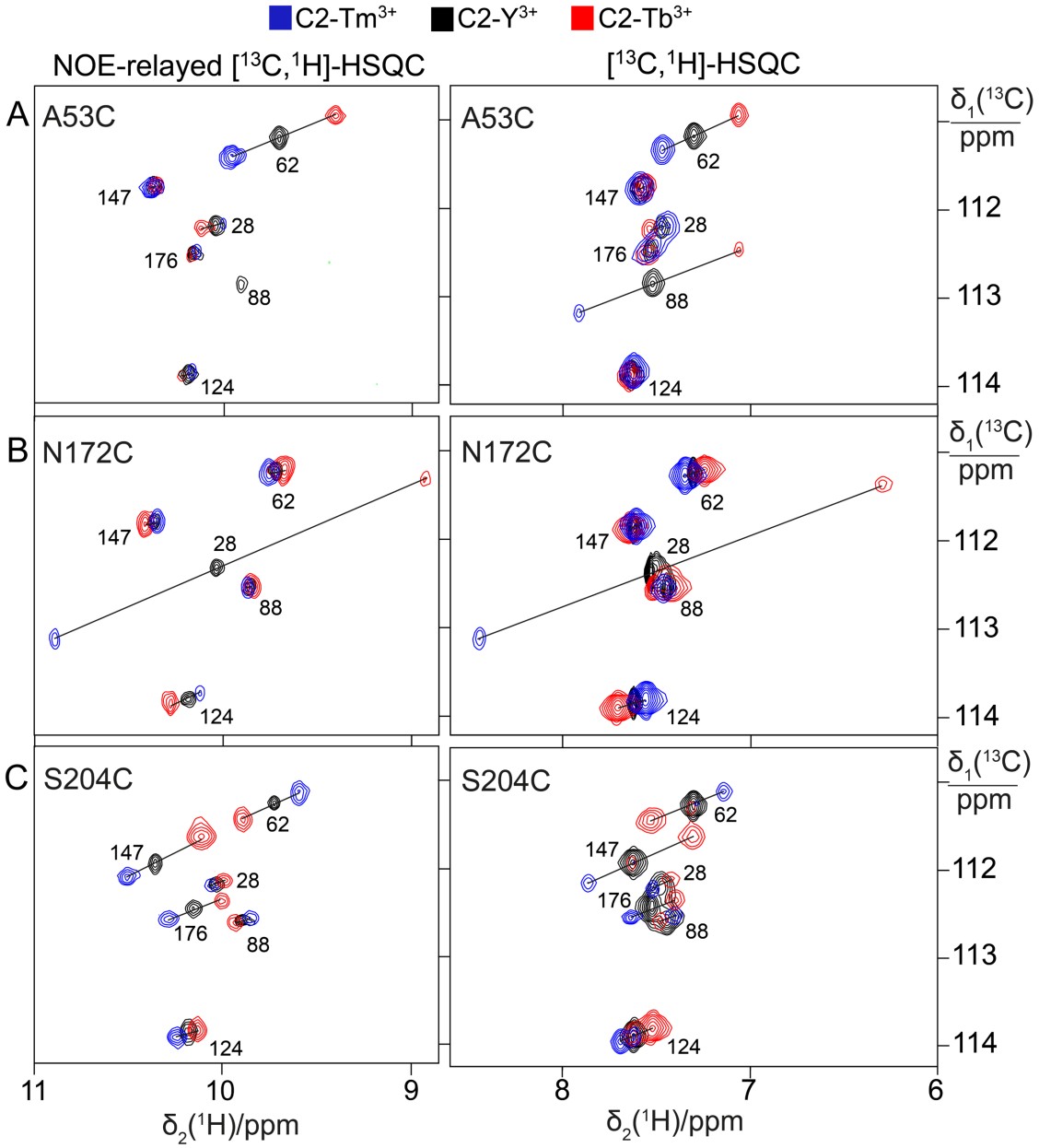


**Figure 2.** PCSs observed in $^{13}C$-$^{1}H$ correlation spectra of 0.4 mM solutions of IMP-1 mutants tagged with C2-Ln$^{3+}$ tags and containing selectively isotope-labelled tryptophan produced from 7-$^{13}C$-indole deuterated in the positions 2, 4, 5 and 6. The plots show superimpositions of spectra recorded with diamagnetic (C2-Y$^{3+}$, black) or paramagnetic (C2-Tb$^{3+}$, red; C2-Tm$^{3+}$, blue) tags. All spectra were recorded with spin-state selection in the $^{13}C$-dimension to record the narrow low-field component of each $^{13}C$-doublet. Right panels: [$^{13}C$,$^{1}H$]-HSQC spectra. Left panels: NOE-relayed [$^{13}C$,$^{1}H$]-HSQC spectra (150 ms NOE mixing time) to record the H$^{\varepsilon1}$ resonances of the tryptophan side chains. PCSs are indicated by lines connecting the peaks of paramagnetic and diamagnetic samples. The cross-peaks are assigned with the residue number of the individual tryptophan residues. (A) Mutant A53C. (B) Mutant N172C. (C) Mutant

S204C.

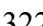

Figure 3. Effect of the presence of L-captopril on the PCSs observed in $^{13}C$-$^1H$ correlation spectra of 0.4 mM solutions of IMP-1 mutants. Protein preparations and experimental parameters were the same as in Fig. 2. Spectra recorded with diamagnetic (C2-Y$^{3+}$, black) or paramagnetic (C2-Tb$^{3+}$, red; C2-Tm$^{3+}$, blue) tags are superimposed. Right column: [$^{13}C$,$^1H$]-HSQC spectra. Left column: NOE-relayed [$^{13}C$,$^1H$]-HSQC spectra recorded with 150 ms NOE mixing time. Stars mark cross-peaks of species putatively attributed to protein degradation. (A) Mutant A53C. (B) Mutant N172C. (C) Mutant S204C.

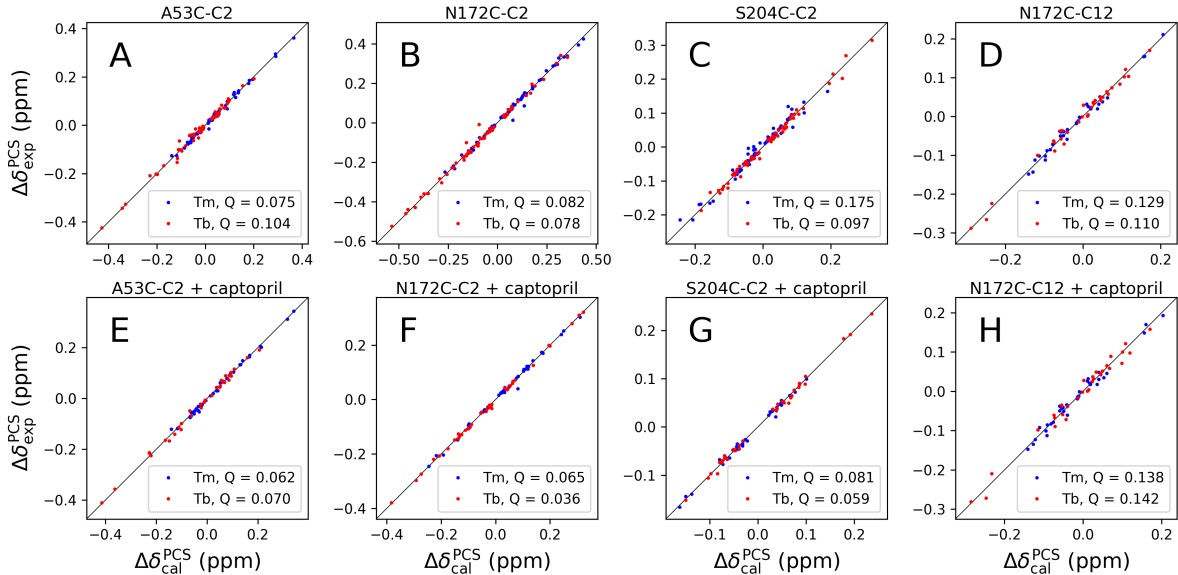

**Figure 4.** Correlations between back-calculated and experimental $^1$H PCSs measured of backbone amides of IMP-1 with C2 tags at three different sites (positions 53, 172 and 204) and the C12 tag in position 172. Red and blue data points correspond to the PCS data obtained with Tb$^{3+}$ and Tm$^{3+}$ tags, respectively. (A) Mutant A53C with C2 tag. (B) Mutant N172C with C2 tag. (C) Mutant S204C with C2 tag. (D) Mutant N172C with C12 tag. (E) Same as (A) but in the presence of captopril. (F) Same as (B) but in the presence of captopril. (G) Same as (C) but in the presence of captopril. (H) Same as (D) but in the presence of captopril. PCS data in (A)–(D) were used to fit $\Delta\chi$ tensors to the structure 5EV6. PCS data in (E)–(F) were used to fit $\Delta\chi$ tensors to the structure 4C1F.

## 3.3 Determining the localisation spaces of tryptophan sidechains

The $\Delta\chi$ tensors determined of backbone amides not only enabled the resonance assignment of the tryptophan sidechains by comparing back-calculated with experimental PCSs, but also allowed translation of the indole PCSs into restraints that define the locations of the tryptophan H$^{\zeta2}$ and H$^{\varepsilon1}$ atoms with respect to the rest of the protein. The concept of localising nuclear spins by PCSs that are generated by lanthanoid tags at different sites is well-established (see, e.g., Yagi et al., 2013; Lescanne et al., 2018; Zimmermann et al., 2019). It can be visualised by representing each PCS restraint by the corresponding PCS isosurface, which comprises all points in space where this PCS value is generated by the $\Delta\chi$ tensor (Fig. 5). With PCS restraints from two different metal sites, the intersection between the respective isosurfaces defines a line. The intersection of this line with the PCS isosurface from a third $\Delta\chi$ tensor defines two points. While a fourth $\Delta\chi$ tensor could unambiguously produce a single solution, a fourth tensor

may not be required if one of these two points is incompatible with the covalent structure of
the protein. In favourable circumstances, the constraints imposed by the covalent structure may
even allow the accurate positioning of nuclear spins by PCSs generated from only two different
$\Delta\chi$ tensors (Pearce et al., 2017). Therefore, the present study was successful with only three
different tagging sites. Figure S13 illustrates the concept for the Trp28 $H^{\varepsilon 1}$ atom.

The spatial definition of the intersection point defined by the PCS isosurfaces depends

on the experimental uncertainties in a non-isotropic way, as the PCS isosurfaces rarely intersect
in an orthogonal manner and the PCS gradients differ for each $\Delta\chi$ tensor. To capture a
localisation space, which allows for the experimental uncertainty in the measured PCS data
and fitted $\Delta\chi$ tensors, we mapped the spatial field of root-mean-squared deviations (RMSD)
between experimental and calculated PCS values and defined the boundary of the localisation
space by a maximal RMSD value. In addition, uncertainties in the $\Delta\chi$ tensors were propagated
by averaging over the results from 20 $\Delta\chi$-tensor fits performed with random omission of 20 %
of the backbone PCS data. In the present work, the routine for defining the localisation space
was implemented as a script in the software Paramagpy (Orton et al., 2020). Figure 6 shows
the resulting localisation spaces for the $H^{\varepsilon 1}$ and $H^{\zeta 2}$ atoms of Trp28, using the PCS data
obtained for the three cysteine mutants A53C, N172C and S204C with the C2-$Tb^{3+}$ tag as well
as the N172C mutant with the C12-$Tb^{3+}$ tag.

The localisation spaces found for the $H^{\varepsilon 1}$ and $H^{\zeta 2}$ atoms of Trp28 were clearly different.

Furthermore, the distance between them corresponded closely to the distance expected from
the chemical structure of the indole ring (2.9 Å). The irregular shapes of the localisation spaces
displayed in Fig. 6 purely reflect the relative geometry of the intersecting PCS isosurfaces and
do not take into account any dynamic flexibility of the L3 loop or protein structure. In
particular, the relevant PCS isosurfaces associated with the C2 tag at sites N172C and S204C
intersect at a shallow angle, which leads to the elongated shape of the localisation space for the
Trp28 $H^{\zeta 2}$ atom (Fig. S13). For the nitrogen-bound $H^{\varepsilon 1}$ atom, the localisation space was
restricted further by the additional data obtained with the C12 tag at site N172C (Fig. 6).
Calculating the localisation spaces from the $Tm^{3+}$ data yielded very similar results (Fig. S14).
The agreement of the localisation spaces of Trp28 with chain A of the previously published
crystal structure 5EV6 is excellent and they are clearly incompatible with the conformations
observed in chain C of the same structure or in the structure 1DDK (Fig. 1A).

Due to close proximity to the C2 tags in the N172C mutant, the largest PCSs were

observed for Trp28 $H^{\varepsilon 1}$ but, in the absence of captopril, their exact magnitude appeared about
0.3 ppm smaller in the [$^{15}$N,$^1$H]-HSQC (Fig. S5b) than the NOE-relayed [$^{13}$C,$^1$H]-HSQC (Fig.
2B) spectrum. The centre of the localisation space of Trp28 H$^{\varepsilon 1}$ moved to a slightly more open
L3 loop conformation when using the smaller PCS detected in the [$^{15}$N,$^1$H]-HSQC spectrum
of the N172C mutant labelled with the C2-Tb$^{3+}$ tag. The space still encompassed the
coordinates observed in the structure 5EV6, limiting the significance of this difference in PCS.

None of the minor additional cross-peaks observed in any of the sample preparations

could be attributed to alternative conformations of Trp28 either. In particular, the most extreme
conformation observed in the crystal structure 1DDK (green in Fig. 1) predicts PCSs > 1 ppm
for Trp28 H$^{\varepsilon 1}$ in the mutant N172C with C2 tags, but we observed no PCS of this magnitude
for any of the unassigned peaks.


**3.4 Defining the localisation space with one versus two lanthanoid ions in the same tag**
**and at the same site**

Unexpectedly, determining separate localisation spaces from the Tm$^{3+}$ and Tb$^{3+}$ datasets
yielded more plausible results than when both datasets were used simultaneously. Careful
inspection showed that the close alignment of the $\Delta\chi$ tensors of the Tm$^{3+}$ and Tb$^{3+}$ data resulted
in particularly shallow intersection angles of the respective PCS isosurfaces. In calculating the
localisation space of Trp28, the PCS isosurfaces arising from the N172C mutant carried by far
the greatest weight as this site is closer to residue 28 than the sites 53 and 204. Therefore, the
Tm$^{3+}$ and Tb$^{3+}$ data from the N172C mutant dominated the PCS RMSD calculation and the
intersection between the associated isosurfaces pulled the final localisation space to a
structurally implausible location, which was unstable with respect to small perturbations in $\Delta\chi$-
tensor orientations associated with the tensors at site 172. In contrast, considering the Tm$^{3+}$ and
Tb$^{3+}$ datasets separately allowed the localisation spaces to be determined by the intersections
with PCS isosurfaces from the other sites. The resulting localisation spaces consistently were
compatible with crystal structures.

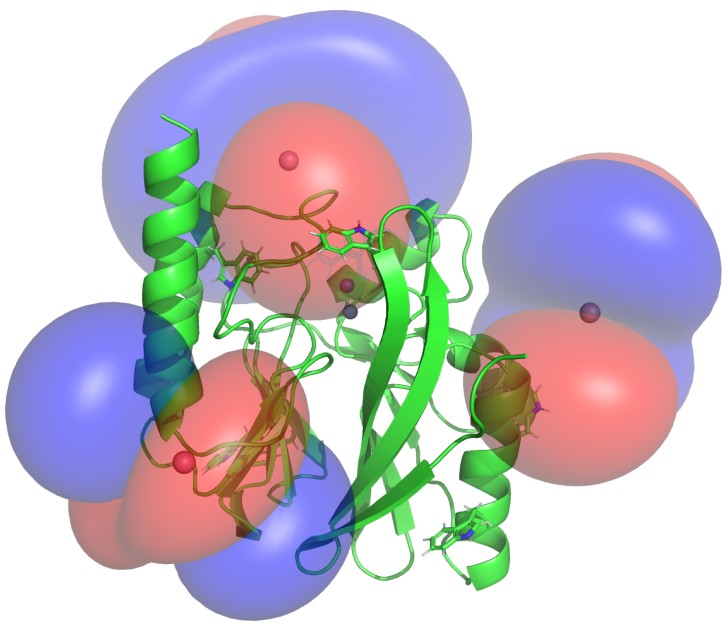


**Figure 5.** PCS isosurfaces of the IMP-1 mutants A53C, N172C and S204C plotted on the crystal structure 5EV6. The respective $\Delta\chi$ tensors were determined from the $^1$H PCSs measured of backbone amides. Blue/red isosurfaces correspond to PCSs of +/-1.0 ppm, respectively, generated with C2-Tb$^{3+}$ tags.

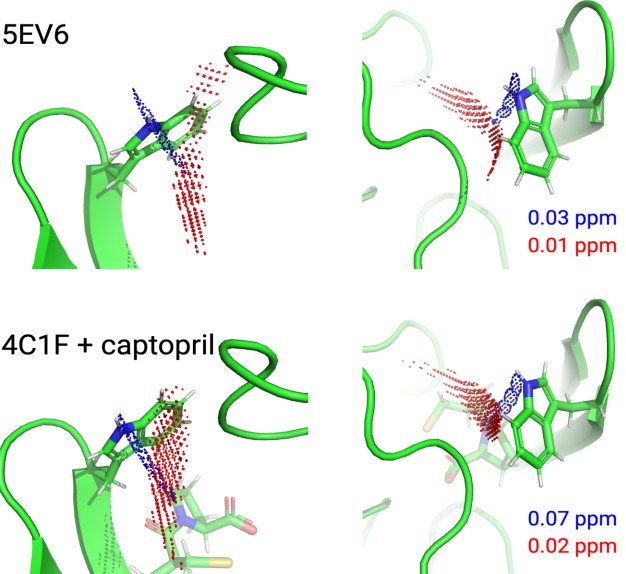

**Figure 6.** Localisation space of the sidechain of Trp28 defined by the PCSs from tags in the IMP-1 mutants A53C, N172C and S204C. The left and right panels display the same results in two different orientations. Red and blue points outline localisation spaces determined for the H$^{\zeta 2}$ and H$^{\varepsilon 1}$ atoms, respectively. The localisation space of the H$^{\zeta 2}$ atom was defined by the PCSs and $\Delta\chi$ tensors determined for the Tb$^{3+}$-loaded C2 tags, while the localisation space of the H$^{\varepsilon 1}$ atom was restricted by additional data obtained with C12-Tb$^{3+}$ tag at site N172C. The

boundaries of the respective localisation spaces displayed are defined by the PCS RMSD values
indicated in ppm. The top panel depicts the localisation spaces determined for the free protein
plotted on chain A of the crystal structure 5EV6 depicted in two different orientations. The
lower panel depicts the localisation spaces determined in the presence of captopril plotted on
chain A of the crystal structure 4C1F.

**3.5 L3 loop conformation in the presence of L-captopril**
Figure 6 shows that, within the uncertainty of the experiments, the localisation space of the
indole sidechain of Trp28 is invariant with respect to the presence or absence of captopril.
Conservation of the L3 loop conformation with and without inhibitor is supported by the close
similarity in all the PCSs observed for Trp28 in the NOE-relayed [$^{13}$C,$^1$H]-HSQC spectra (Fig.
2 and 3). In the [$^1$H,$^{15}$N]-HSQC spectra of the mutant N172C with C2 tag, however, the PCSs
observed for Trp28 H$^{\varepsilon 1}$ appeared somewhat smaller without than with captopril (Fig. S5b). As
the PCSs of backbone amides were very similar in the absence and presence of the inhibitor
(Fig. S12), this difference in PCS suggests a change in L3 loop conformation, contradicting the
observations made with the selectively $^{13}$C-labelled samples. As discussed above, using the
smaller PCS of Trp28 H$^{\varepsilon 1}$ did not sufficiently change its localisation space in the free protein
to render it incompatible with the coordinates of the structure 5EV6. Therefore, as far as the
data of the $^{15}$N-labelled samples indicate a conformational change of the L3 loop between the
free and bound state, it is small. We attribute the differences in PCSs observed between the
selectively $^{13}$C-labelled and uniformly $^{15}$N-labelled samples to differences in sample
preparation of unknown origin, which are also reflected by different numbers of weak
unassigned cross-peaks (Figs 2, 3, S5 and S6).
The cross-peak intensities of the Trp28 sidechain resonances are relatively weak
compared with those of the other tryptophan sidechains, suggesting that Trp28 is subject to
dynamics that broaden its resonances. Its cross-peaks appeared slightly weaker in the presence
than in the absence of inhibitor (Fig. 2 and 3), suggesting a change in dynamics caused by the
inhibitor binding. Previous NMR studies of metallo-β-lactamases reported faster $R_2(^{15}$N)
relaxation rates of the L3-loop tryptophan sidechain in the presence than in the absence of
inhibitor, which was attributed to dampened dynamics (Huntley et al., 2000; Softley et al.,
2020). In the presence of dynamics, the localisation spaces determined in the present work
must be considered averages that do not report on the amplitude or direction of motions.

**3.6 Localisation spaces of tryptophan side chains other than Trp28**

As the tagging sites had been designed to analyse the conformation of the L3 loop, they were positioned at similar distances from the L3 loop and therefore not optimal for determining localisation spaces of the other tryptophan residues. Nonetheless, clear differences were observed in the PCSs of the $H^{\xi 2}$ and $H^{\varepsilon 1}$ atoms (Fig. 2), allowing the separation of the respective localisation spaces, which also proved to be in excellent agreement with the conformations of the side-chain indoles of Trp62, Trp124 and Trp147 as found in the crystal structure (Fig. S15), whereas the data were insufficient to determine the sidechain conformation of Trp176.

**4 Discussion**

The L3 loop of metallo-β-lactamases is known to be flexible and, in the specific case of IMP-1, significantly assists in substrate binding and enzymatic activity (Moali et al., 2003). As the substrate is sandwiched between the di-zinc site and the L3 loop, it is tempting to think that the loop opens up for substrate binding and product release while it may be closed during the enzymatic reaction to hold the substrate and reaction intermediate in place. In contrast, some of the conformations observed in crystal structures of IMP-1 obtained in the presence and absence of the inhibitor L-captopril, revealed the loop in almost identical conformations (Brem et al., 2016). This observation is inconclusive, however, as the L3 loop forms more extensive intermolecular contacts with neighbouring protein molecules in the crystal lattice than intramolecular contacts. In addition, other crystal structures observed the loop to move by almost 3 Å in response to a different inhibitor (Concha et al., 2000). This prompted us to probe its actual location in the absence of crystal packing forces in solution, a task which is difficult to tackle by traditional NMR spectroscopic methods that rely on short-range NOEs.

Our results show that by furnishing IMP-1 with paramagnetic lanthanoid tags, the coordinates of the indole sidechain of Trp28, which is a key residue near the tip of the loop, can be determined with remarkable accuracy even in the free protein, where the available crystal structures position the L3 loop in a conformation without any direct contacts with the core of the protein. Indeed, the localisation space identified by the NMR data of the free protein proved to be sufficiently well-defined to discriminate between different crystal structures of IMP-1, as well as between different chains in the same asymmetric crystal unit. For example, the sidechain orientation of Trp28 observed in $[Fe^{3+},Zn^{2+}]$-IMP-1 (4UAM; Carruthers et al., 2014) proved to be in poor agreement with the PCS data, whereas the data were in full agreement with chain A in the structure 5EV6 of $[Zn^{2+},Zn^{2+}]$-IMP-1 without inhibitor

(Hinchliffe et al., 2016) and chain A in the structure 4C1F with bound L-captopril (Brem et al.,
2016). This highlights the outstanding capacity of PCSs to assess small conformational
differences.

The approach of using PCSs for local structure determination is particularly appealing

in the case of difficult proteins such as IMP-1, where the sequence-specific NMR resonance
assignments are incomplete due to line-broadening attributable to motions in the µs–ms time
range and additional signals are observed that either stem from protein degradation, misfolding
or alternative conformations in slow exchange with the main structure. Notably, all information
required to establish the $\Delta\chi$ tensors could be obtained from resolved cross-peaks observed in
sensitive [$^{15}$N,$^{1}$H]-HSQC spectra. Similarly, the localisation information of the tryptophan
sidechains could be obtained from sensitive $^{13}$C-$^{1}$H and $^{15}$N-$^{1}$H correlation spectra. Positioning
the lanthanoid tags relatively far from the substrate binding site avoided direct interference
with the binding loop structure.

In the face of additional signals from minor species, site-selective $^{13}$C-labelling of the

tryptophan sidechains was particularly helpful for simplifying the [$^{13}$C,$^{1}$H]-HSQC spectra.
Gratifyingly, this could be achieved by providing suitably labelled indole without having to
synthesise the full amino acid (Maleckis et al., 2021).

It has been pointed out previously that the accuracy with which localisation spaces can

be determined is best when PCS isosurfaces intersect in an orthogonal manner (Pintacuda et
al., 2006; Lescanne et al., 2018; Zimmermann et al., 2019). In the present work, we found that,
counterintuitively, the provision of additional data can considerably degrade the accuracy of
the localisation space. This effect arises when PCS isosurfaces intersect at a shallow angle, as
the location of these intersections becomes very sensitive with regard to small errors in the
relative orientations of the underpinning $\Delta\chi$ tensors. Shallow intersection angles of PCS
isosurfaces are common, when two PCS datasets are from tags and tagging sites that differ only
in the identity of the paramagnetic metal ion in the tag. This situation commonly generates $\Delta\chi$
tensors of different magnitude and sign, but closely similar orientation (Bertini et al., 2001; Su
et al., 2008; Keizers et al., 2008; Man et al., 2010; Graham et al., 2011; Joss et al., 2018;
Zimmermann et al., 2019). Therefore, while the use of Tm$^{3+}$ and Tb$^{3+}$ tags is helpful for
assigning the cross-peaks in the paramagnetic state, more robust results are obtained by using
only one of these data sets for calculating the localisation space. Good localisation spaces were
thus obtained by using only PCSs measured for Tb$^{3+}$ tags (Fig. 6) or only PCSs measured for
Tm$^{3+}$ tags (Fig. S13). In contrast, however, very different tags attached at the same site, such
as the C2 and C12 tags installed in the mutant N172C, produced independent $\Delta\chi$-tensor
orientations and therefore contributed positively to localising the Trp28 H$^{\varepsilon 1}$ atom.
In principle it is inappropriate to explain a set of PCSs by a single $\Delta\chi$ tensor, if they are
generated by a lanthanoid tag attached via a flexible linker, which positions the lanthanide ions
at variable coordinates relative to the protein. In this situation, fitting a single $\Delta\chi$ tensor
amounts to an approximation. The effective $\Delta\chi$ tensors obtained in this way, however, can
fulfill the PCSs remarkably well (Shishmarev and Otting, 2013), as illustrated by the low $Q$
factors obtained in this work (Fig. 4), and the localisation spaces obtained for the tryptophan
sidechains are correspondingly well defined.
The accuracy, with which localisation spaces can be determined, further depends on the
accuracy with which PCSs can be measured (which critically depends on the reproducibility of
the sample conditions between the paramagnetic and diamagnetic states), the accuracy of the
protein structure used to fit the $\Delta\chi$ tensors and the angle with which PCS isosurfaces of different
tensors intersect. To take into account the uncertainties associated with the PCS isosurfaces, it
is useful to think of each of them individually as a shell of a certain thickness (rather than a
surface) that represents a compatible localisation space. Two shells of a given thickness share
a smaller common space if they intersect orthogonally than if they intersect at a shallow angle.
The present work employed $^{1}$H PCSs only, although PCSs were also observed in the
indirect dimensions of the [$^{13}$C,$^{1}$H]-HSQC and [$^{15}$N,$^{1}$H]-HSQC spectra. We made this choice
because the paramagnetic tags give rise to weak molecular alignments in the magnetic field,
which result in residual anisotropic chemical shifts (RACS). The effect is unimportant for $^{1}$H
spins but significant for nuclear spins with large chemical shift anisotropy (CSA) tensors such
as backbone nitrogens and aromatic carbons. Correcting for the RACS effect is possible with
prior knowledge of the CSA tensors and bond orientations (John et al., 2005). We therefore
chose not to measure PCSs of the heteronuclear spins in favour of improving sensitivity by
accepting a lower spectral resolution in the indirect dimensions.
Finally, the C12 tag was designed specifically with the intent to produce a more rigid
tether to the protein than the C2 tag, but this did not result in larger $\Delta\chi$ tensors (Table S7) and
the NMR spectra of IMP-1 N172C displayed more heterogeneity with the C12 than the C2 tag,
suggesting that the shorter and more rigid tether combined with the fairly high molecular
weight of the cyclen-lanthanoid complex may have perturbed the protein structure to some
degree.

## 5 Conclusion

The current work illustrates how $\Delta\chi$ tensors from paramagnetic lanthanoid ion tags installed at three different sites of the protein can be used to probe the conformation of a selected site in solution in unprecedented detail, provided the structure of most of the protein is known with high accuracy to allow fitting effective $\Delta\chi$ tensors of high predictive value. Importantly, however, the method is easily compromised, if two PCS isosurfaces intersect at a shallow angle as, in this situation, inaccuracies in $\Delta\chi$ tensor determinations have an outsized effect on positioning the localisation spaces defined by the PCSs. Therefore, improved results were obtained by not combining data from different metal ions bound to otherwise identical tags and tagging sites. In the present work, simplifying the NMR spectrum of tryptophan residues by site-selective isotope labelling proved to be of great value for sufficiently improving the spectral resolution to allow assigning the labelled resonances solely from PCSs and PREs. The strategy opens a path to detailed structural investigations of proteins of limited stability like IMP-1, for which complete assignments of the NMR spectrum are difficult to obtain.

**Code and data availability.** NMR spectra and pulse programs are available at https://doi.org/10.5281/zenodo.5518294. The script for calculating localisation spaces is available at https://doi.org/10.5281/zenodo.3594568 and from the GitHub site of Paramagpy.

**Supplement.** The supplement related to this article is available online at: https://doi.org/...

**Author contributions.** GO initiated the project and edited the final version of the manuscript. HWO wrote NMR pulse programs and software to calculate localisation spaces and performed the $\Delta\chi$ tensor and structure analysis. IDH made labelled protein samples, recorded and assigned NMR spectra, measured PCSs and wrote the first version of the manuscript. AM synthesised the isotope-labelled indole. SJ made [15]N-labelled protein mutants with C2 tags and assigned PCSs of backbone amides. MS synthesized C2 tags with different lanthanoid ions. CB, LT and SB synthesized C12 tags with different lanthanoid ions.

**Competing interests.** The authors declare that they have no conflict of interest.

**Financial support.** GO thanks the Australian Research Council for a Laureate Fellowship (grant no. FL170100019) and project funding through the Centre of Excellence for Innovations

in Peptide and Protein Science, Australian Research Council (grant no. CE200100012). AM
thanks the European Regional Development Fund (ERDF) for funding (PostDoc project
No. 1.1.1.2/VIAA/2/18/381).

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
