# Peer review of "Localising individual atoms of tryptophan side chains in the metallo- $\beta$ -lactamase IMP-1"

_Magnetic Resonance, 2021_

## Author Comment (AC1)

(1) The triangulation approach of using the intersections of three PCS isotherms has been reported before in other papers (e.g. J Biomol NMR 71, 27, 2018), so it is not clear why the current approach is not compared with published methods.

There seems to be a typo in the page number - we presume that the reference referred to is J Biomol NMR 71, 271, 2018 (Lescanne et al.), which is indeed relevant.
Using different paramagnetic metal ions and paramagnetic metal tags attached to proteins at two or more sites (one at a time) is a well-established strategy for pinpointing the location of protein sidechains, bound ligand molecules or proteins, as well as for 3D structure determinations of proteins. We propose to include the following (broadly representative though still incomplete) list of references in the introduction.

For side chain conformations:
Pearce, B. J. G. Jabar, S., Loh, C.-T., Szabo, M., Graham, B. and Otting, G.: Structure restraints from heteronuclear pseudocontact shifts generated by lanthanide tags at two different sites, J. Biomol. NMR, 68, 19–32, https://doi.org/10.1007/s10858-017-0111-z, 2017.

Lescanne, M., Ahuja, P., Blok, A., Timmer, M., Akerud, T. and Ubbink, M. Methyl group reorientation under ligand binding probed by pseudocontact shifts, J. Biomol. NMR, 71, 275–285, https://doi.org/10.1007/s10858-018-0190-5, 2018.

For protein-ligand complexes:
Guan, J.-Y., Keizers, P. H. J., Liu, W.-M., Löhr, F., Skinner, S. P., Heeneman, E. A., Schwalbe, H., Ubbink, M. and Siegal, G.: Small-molecule binding sites on proteins established by paramagnetic NMR spectroscopy, J. Am. Chem. Soc., 135, 5859–5868, https://doi.org/10.1021/ja401323m, 2013.

Chen, W.-N., Nitsche, C., Pilla, K. B., Graham, B., Huber, T., Klein, C. D. and Otting, G.: Sensitive NMR approach for determining the binding mode of tightly binding ligand molecules to protein targets, J. Am. Chem. Soc., 138, 4539–4546, https://doi.org/10.1021/jacs.6b00416, 2016.

Zimmermann, K., Joss, D., Müntener, T., Nogueira, E. S., Schäfer, M., Knörr, L., Monnard, F. W. and Häussinger, D.: Localization of ligands within human carbonic anhydrase II using $^{19}$F pseudocontact shift analysis, Chem. Sci., 10, 5064–5072, https://doi.org/10.1039/c8sc05683h, 2019.

For protein-protein complexes:
Pintacuda, G., Park, A. Y., Keniry, M. A., Dixon, N. E. and Otting, G.: Lanthanide labeling offers fast NMR approach to 3D structure determinations of protein-protein complexes, J. Am. Chem. Soc., 128, 3696–3702, https://doi.org/10.1021/ja057008z, 2006.

Keizers, P. H. J., Mersinli, B., Reinle, W., Donauer, J., Hiruma, Y., Hannemann, F., Overhand, M., Bernhardt, R. and Ubbink, M.: A solution model of the complex formed by adrenodoxin and adrenodoxin reductase determined by paramagnetic NMR spectroscopy, Biochemistry, 49, 6846–6855, https://doi.org/10.1021/bi100598f, 2010.

de la Cruz, L., Nguyen, T. H. D., Ozawa, K., Shin, J., Graham, B., Huber, T. and Otting, G.: Binding of low molecular weight inhibitors promotes large conformational changes in the dengue virus NS2b-NS3 protease: fold analysis by pseudocontact shifts, J. Am. Chem. Soc., 133, 19205–19215, https://doi.org/10.1021/ja208435s, 2011.

Kobashigawa, Y., Saio, T., Ushio, M., Sekiguchi, M., Yokochi, M., Ogura, K. and Inagaki, F.: Convenient method for resolving degeneracies due to symmetry of the magnetic susceptibility tensor and its application to pseudo contact shift-based protein-protein complex structure determination, J. Biomol. NMR, 53, 53–63, https://doi.org/10.1007/s10858-012-9623-8, 2012.

Brewer, K. D., Bacaj, T., Cavalli, A., Camilloni, C., Swarbrick, J. D., Liu, J., Zhou, A., Zhou, P., Barlow, N., Xu, J., Seven, A. B., Prinslow, E. A., Voleti, R., Häussinger, D., Bonvin, A. M. J. J., Tomchick, D. R., Vendruscolo, M., Graham, B., Südhof and T. C., Rizo, J.: Dynamic binding mode of a synaptotagmin-1-SNARE complex in solution. Nat. Struct. Mol. Biol., 22, 555–564, https://doi.org/10.1038/nsmb.3035, 2015.

For 3D protein structure determination:
Yagi, H., Pilla, K. B., Maleckis, A., Graham, B., Huber, T. and Otting, G.: Three-dimensional protein fold determination from backbone amide pseudocontact shifts generated by lanthanide tags at multiple sites, Structure, 21, 883–890, https://doi.org/10.1016/j.str.2013.04.001, 2013.

Crick, D. J., Wang, J. X., Graham, B., Swarbrick, J. D., Mott, H. R. and Nietlispach, D.: Integral membrane protein structure determination using pseudocontact shifts. J. Biomol. NMR, 61, 197–207, https://doi.org/10.1007/s10858-015-9899-6, 2015.

Pilla, K. B., Otting, G. and Huber, T.: Protein structure determination by assembling super-secondary structure motifs using pseudocontact shifts, Structure, 25, 559–568, https://doi.org/10.1016/j.str.2017.01.011, 2017.

(2) In the discussion (l. 479 – 486), the point of not using different metals in the same tag but multiple orthogonal sites has been made by other studies, so references are required there.

We agree that the discussion needs to acknowledge previous findings regarding the importance of isosurfaces intersecting in an orthogonal manner. The comment raises an interesting question: are the tensor axes also aligned, when comparing tensors generated by paramagnetic lanthanoid ions and transition metal ions such as Co(II)? We remember having observed this once:

Man, B., Su, X.-C., Liang, H., Simonsen, S., Huber, T., Messerle, B.A. and Otting, G.: 3-Mercapto-2,6-pyridinedicarboxylic acid, a small lanthanide-binding tag for protein studies by NMR spectroscopy, Chem. Eur. J., 16, 3827–3832, https://doi.org/10.1002/chem.200902904, 2010.

but there may be more examples.

We suggest the following new paragraph:

It has been pointed out previously that the accuracy with which localisation spaces can be determined is best when PCS isosurfaces intersect in an orthogonal manner (Pintacuda et al., 2006; Lescanne et al., 2018; Zimmermann et al., 2021). In the present work, we found that, counterintuitively, the provision of additional data can considerably degrade the accuracy of the localisation space. This effect arises when PCS isosurfaces intersect at a shallow angle, as the location of these intersections becomes very sensitive with regard to small errors in the relative orientations of the underpinning Dc tensors. Shallow intersection angles of PCS isosurfaces routinely occur, when two PCS datasets are from tags and tagging sites that differ only in the identity of the paramagnetic metal ion in the tag. This situation commonly generates Dc tensors of different magnitude and sign, but closely similar orientation (Bertini et al., 2001; Su et al., 2008; Keizers et al., 2008; Man et al., 2010; Graham et al., 2011; Joss et al., 2018; Zimmermann et al., 2021). Therefore, while the use of $Tm^{3+}$ and $Tb^{3+}$ tags is helpful for assigning the cross-peaks in the paramagnetic state, it is safer to use only one of these data sets for calculating the localisation space. Good localisation spaces were thus obtained by using only PCSs measured for $Tb^{3+}$ tags (Fig. 6) or only PCSs measured for $Tm^{3+}$ tags (Fig. S12). In contrast, however, very different tags attached at the same site, such as the C2 and C12 tags installed in the mutant N172C, produced independent Dc-tensor orientations and therefore contributed positively to localising the Trp28 $H^{e1}$ atom.

(3) In line 246 it is mentioned that double peaks are observed for the Trp NHe groups. That could mean that the indoles are in different conformations in slow exchange. It is not discussed whether these could be the other conformations observed in the crystal structures. Could the PCS analysis be done for these minor peaks to exclude that possibility? In that case this work only yields the position of the major form, not the only form.

We do not understand the origin of the minor species and could not assign these weak cross-peaks sequence-specifically, also not with the help of PCSs. In previous work, we discovered that IMP-1 readily installs a Fe(III) ion one of the Zn(II) sites (Carruthers et al., 2014), which explained some of the additional peaks observed in early preparations. Our current protein purification protocol carefully excludes iron. As the cross-peaks of the remaining minor species varied in intensity between different

sample preparations and following long NMR measurements, we believe that they arise from some process of sample degradation.

(4) The mass spectrometry shown in Fig. S2 and the yields mentioned give rise to questions. In line 204 the efficiency of 90% is mentioned. However, using the information in the caption of Fig. S2, a different result is suggested: The masses in the figure are about 9 Da lower than expected for 100% labelling (given the mentioned masses), which 25% of the expected 36 Da extra (in the caption it says +6 Da/Trp), so labelling efficiency would be 75%. However, after converting the indole to tryptophan, one deuteron is removed, so the expected mass increase is 4XD + 1 13C = 5 Da per Trp, not 6. That would result in a labelling of 90%, agreeing with the main text, but suggesting that the masses mentioned in the caption are too high. Please check.

Thank you for alerting us to the error. Following double-checking, the calculated masses for the spectra of Figure S2a-c should have been reported as 25234.88 Da, 25191.86 Da and 25218.88 Da, respectively, in agreement with 90 % incorporation efficiency of the isotope-labelled indole. We also noticed that the spectra shown in Figure 2d and f had accidentally been swapped and that the tag used increased the mass by 858 Da (not 832 Da).

(5) In section 2.2, mention the protein concentration(s) used for the NMR samples and indicate the tube type (3 mm, 5 mm, Shigemi), to know how much sample was used. Also mention the protein concentration in the captions of the NMR spectra figures in the supplementary material.

The spectra were recorded using 3 mm tubes. The requested information will be provided in Section 2.2. The protein concentrations were already given in the captions of the overview NMR spectra shown in the supplementary material.

(6) l. 213, how were the assignments for the diamagnetic protein obtained? If from previous reports, give the reference and the BMRB entry.

The assignments of the diamagnetic protein were already released in 2014 in BMRB entry 25063. We will refer to this entry in Section 2.3.

(7) Thank you for pointing out these oversights, which we will fix in the revised version.

---

## Author Response (AR1)

In response to the evaluation by Marcellus Ubbink:

(1) The triangulation approach of using the intersections of three PCS isotherms has been reported before in other papers (e.g. J Biomol NMR 71, 27, 2018), so it is not clear why the current approach is not compared with published methods.

> Response: The reference referred to (Lescanne et al., J Biomol NMR 71, 271, 2018) is indeed relevant, as are several others, the most important ones we have now attempted to include in the list of references (cited on lines 104-110). We now explicitly state that using different paramagnetic metal ions and paramagnetic metal tags attached to proteins at two or more sites (one at a time) is a well-established strategy for pinpointing the location of protein sidechains, bound ligand molecules or proteins, as well as for 3D structure determinations of proteins. We also made sure to avoid any impression of a claim that the strategy per se is novel. Instead, our contribution is to (i) illustrate the accuracy with which structural details in a solvent-exposed loop can be elucidated in this way and (ii) highlight the finding that PCSs from two paramagnetic metal ions (rather than one) in the same tag at the same site can produce a worse result than reducing the PCS data set to include only data from a single paramagnetic metal ion per tag and tagging site (now pointed out more clearly in the Discussion and Conclusion sections).

(2) In the discussion (l. 479 – 486), the point of not using different metals in the same tag but multiple orthogonal sites has been made by other studies, so references are required there.

> Response: We now cite previous work pointing out the importance of isosurfaces intersecting in an orthogonal manner more comprehensively (Section 3.3 and the Discussion, lines 498-513). Nonetheless, some success has been had in the past with different metal ions in the same site, and this is now discussed too. We believe that it has not been reported previously that more data can give objectively worse results.

(3) In line 246 it is mentioned that double peaks are observed for the Trp NHe groups. That could mean that the indoles are in different conformations in slow exchange. It is not discussed whether these could be the other conformations observed in the crystal structures. Could the PCS analysis be done for these minor peaks to exclude that possibility? In that case this work only yields the position of the major form, not the only form.

> Response: We do not understand the origin of the minor species and could not assign these weak cross-peaks sequence-specifically, also not with the help of PCSs. In previous work, we discovered that IMP-1 readily installs a Fe(III) ion one of the Zn(II) sites (Carruthers et al., 2014), which explained some of the additional peaks observed in early preparations (Carruthers 2014). Our current protein purification protocol carefully excludes iron. As the cross-peaks of the remaining minor species varied in intensity between different sample preparations and following long NMR

measurements, we believe that they arise from some process of sample degradation. We now point out that IMP-1 is a protein of limited stability.

(4) The mass spectrometry shown in Fig. S2 and the yields mentioned give rise to questions. In line 204 the efficiency of 90% is mentioned. However, using the information in the caption of Fig. S2, a different result is suggested: The masses in the figure are about 9 Da lower than expected for 100% labelling (given the mentioned masses), which 25% of the expected 36 Da extra (in the caption it says +6 Da/Trp), so labelling efficiency would be 75%. However, after converting the indole to tryptophan, one deuteron is removed, so the expected mass increase is 4XD + 1 13C = 5 Da per Trp, not 6. That would result in a labelling of 90%, agreeing with the main text, but suggesting that the masses mentioned in the caption are too high. Please check.

Response: Unfortunately, the spectra shown in Figure S2d and f had accidentally been swapped, and the mass increase by the tag had been incorrect too. We double-checked all mass spectrometry data carefully and concluded that they do not allow claiming >80 % incorporation efficiency of the isotope-labelled indole (line 218). To allow the reader to check the calculated mass, we specify the exact amino acid sequence of our construct in Section 2.1.1.

(5) In section 2.2, mention the protein concentration(s) used for the NMR samples and indicate the tube type (3 mm, 5 mm, Shigemi), to know how much sample was used. Also mention the protein concentration in the captions of the NMR spectra figures in the supplementary material.

Response: The spectra were recorded using 3 mm tubes. The requested information is now provided in Section 2.2. The protein concentrations were already given in the captions of the overview NMR spectra shown in the supplementary material.

(6) l. 213, how were the assignments for the diamagnetic protein obtained? If from previous reports, give the reference and the BMRB entry.

Response: The assignments of the diamagnetic protein were released in 2014 in BMRB entry 25063. This is now referred to in Section 2.3.

(7) Some supplementary figures have the wrong numbers in the text:
line 165, Fig S1 > S8;
line 170, Fig. S2 > S9;
line 212, Fig. S2-S5, S3-S6 (?)
Tables S1 – S6 are not mentioned in the text.

Response: Thank you for pointing out these oversights, which we corrected in the revised version.

In response to the evaluation by Daniel Häussinger:

1) The procedure of the triangulation by PCS is not new and earlier work by some of the authors and others should be referenced accordingly.

Response: We agree and included a set of references in the introduction (as per our response to comment 1 by Marcellus Ubbink).

2) The finding, that two sets of PCS created at the same tagging site and by the same tag, but different lanthanoids give less accurate results, is not new but certainly remarkable – it would be tempting to quantify this finding by elucidating the "angle score" parameter for these data sets as suggested by Joss et al. (Chem. Sci., **2019**, 10, 5064-5072.)

Response: The angle score parameter is useful. We are working on a more comprehensive comparison of different metrics for the identification of localisation spaces in a separate publication. In the present context, we refer to our response to comment 2 by Marcellus Ubbink.

3) The authors refer only to proton PCS despite the fact, that they obtained also PCS data from $^{15}$N and $^{13}$C – could you comment on that?

Response: We use only $^1$H PCSs because nuclear spins with large CSA tensors display residual anisotropic chemical shift (RACS) effects due to weak paramagnetic alignment of the molecule in the magnetic field. The mixture of alignment and PCS information is cumbersome to disentangle. Furthermore, paramagnetic shifts measured of heteronuclei in the indirect dimension tend to come with greater uncertainties. We now point this out in the Discussion section of the revised version (lines 523-558).

4) The description of the MS instrumentation and conditions is missing.

Response: We now added the information in the supporting information (page S3).

5) The suggested incorporation level of the $^{13}$C-indole is not in accordance with the presented MS spectra in Fig S2, panels a-c.

The figure should be reproduced with better resolution of the individual spectra to allow judgement of the incorporation and a deconvolution of the different isotopomers should be provided.

Response: Some of the data in Figure S2 were swapped by mistake, see our response to point 4 by Marcellus Ubbink. We are not entirely happy with the accuracy of our mass spectrometric measurements, which seem to be no better than 1 Da. We amended the claim of 90 % incorporation rate to 80 % (line 218). Compared to the tallest mass peak, we feel that expanded versions, as provided in our response during the discussion phase, do not offer a more accurate determination of the incorporation rate of the deuterated and 13C-labelled indoles.

6) It would be useful to have the individual metal – cysteine-sulphur distances for each tensor included in table S7

Response: We have provided the information on metal-to-beta-carbon distances in Table S7. The distances vary between 8.2 and 12.1 Å.

7) In the experimental section it is reported that the conjugation of the tags to the protein was performed in the presence of 100 uM $ZnSO_4$ (buffer A), does this not interfere with the cysteines, given the nM Kd values of Zn(Cysteine)$_2$ complexes?

Response: The presence of zinc was necessary as IMP-1 binds traces of iron more tightly (Carruthers et al., Angew. Chem. Int. Ed. 2014, 53, 14268). A single sidechain thiol group doesn't bind zinc very tightly. The solutions contained no free cysteine and there was no evidence for zinc interfering in the ligation reaction.

8) minor typos:
line 142: should read "preparation of each sample"
line 165 should read "Fig. S8"
line 170 should read "Fig. S9"
line 204 c.f. 5)
line 207: "100%" on a deconvoluted mass spectrum is probably stretching the significance a bit – how about "virtually quantitative" or " > 95%"?
line 346: the "of" in "isosurfaces of associated" seems superfluous to me.

Response: We are grateful for the careful evaluation and fix these issues in the revised version. We accept that it is rarely justified to talk of 100 % yield and changed the sentence accordingly (line 221).

---

## Author Response (AR2)

Dear Stephan,

In response to your comments, we have made the following changes. We refer to the original manuscript as ms_orig and the current revised version as ms_current.

1. The major goal of the manuscript is to define the conformation/dynamics around W28 in solution. However, this is not very much worked out. I found the description on lines 401–409 very confusing. Apparently, the PCS of Trp28 Heps1 is scaled down without captopril (Fig. S5b, line 403). Contrary to what is stated on lines 404, 405 I could not see the backbone amide PCS in Fig. S11. Why would this suggest different loop conformations for differently labeled samples (line 406)? In my understanding, the reduced PCS of Trp28 Heps1 in the absence of captopril may very well indicate flexibility. This would also agree with the observed flexibility of the L3 loop and the Trp28 sidechain in the mentioned NMR relaxation study (line 71). The entire discussion on lines 401–409 should be clarified and the comparison to the earlier relaxation data should be made. Eventually this should rather be a key point of the conclusion section.

Response:
The goal was to define the possible change in conformation of an active-site loop of IMP-1 in response to inhibitor binding. As we were unable to pin down conformational changes in an unambiguous manner, we feel that this cannot be a key point of the conclusions. NMR work with this protein is difficult due to the occurrence of peak doubling as well as missing cross-peaks. Missing cross-peaks probably arise from conformational exchange in the μs–ms time regime in parts of the protein. Peak doubling, however, behaves in more mysterious ways, as minor additional peaks of residues near the active site tend to vanish with time. This observation (reported in the PhD thesis by T. J. Carruthers in 2014) was only made for the [ZnZn] complex, but not for the [FeZn] complex, which produces cleaner NMR spectra. We now illustrate minor peaks in the $^{15}$N-HSQC spectrum of the [ZnZn] complex of wild-type IMP-1 selectively labelled with $^{15}$N-tryptophan (Fig. S6). At the same time, sample degradation is an issue. (For example, IMP-1 cannot be stored frozen without irreversible unfolding and precipitation.) As minor peaks arise from different species, do not appear in all spectra (apparent from a comparison of Figs S5 and S6) and were too weak to measure PCSs, we cannot assign and characterize in terms of structure. We feel that an extensive discussion of minor species detracts from the main conclusions of our work, namely the capacity of PCSs to extract structural information for difficult samples, the advantages gained from selectively $^{13}$C-labelled indoles and the effect that localisation spaces can be determined with greater accuracy by omitting data from similar $\Delta\chi$ tensors.

Regarding the scaling down of the PCSs of Trp28 H$^{\varepsilon1}$ in the presence of captopril (line 403 of ms_orig), this effect was not observed in all spectra. It was also small. To recapitulate, we basically have two sets of spectra. Set I comprises the NOE-relayed $^{13}$C-HSQC spectra of $^{13}$C-indole-labelled protein (Figs 2 and 3). Set II comprises the $^{15}$N-HSQC spectra of uniformly $^{15}$N-labelled protein (Figs S5 and S8). The PCSs of the backbone amides, as far as they could be assigned, were practically the same with and without inhibitor (shown in Fig. S12), indicating conservation of the $\Delta\chi$ tensors. This allows attributing changes in PCSs displayed by the side-chain of the tryptophan in the loop of

interest (Trp28 in the L3 loop) to differences in its location rather than changes in $\Delta\chi$ tensor (as described in the first paragraph of Section 3.5).

In the spectra of Set I, the PCSs of Trp28 $H^{\epsilon 1}$ were practically the same with and without inhibitor. In Set II, however, they were slightly smaller in the absence of inhibitor (most apparent in Fig. S5b). As the localisation spaces calculated with either set of PCSs overlapped, the significance of these differences is limited. We now write (lines 478–481 of ms_current): "We attribute the differences in PCSs observed between the selectively $^{13}$C-labelled and uniformly $^{15}$N-labelled samples to differences in sample preparation of unknown origin, which are also reflected by different numbers of weak unassigned cross-peaks (Figs 2, 3, S5 and S6)."

Relaxation studies have never been performed of IMP-1 (only of the related metallo-β-lactamases referenced on line 415 of ms_orig and line 487 of ms_current), but IMP-1 almost certainly behaves the same, i.e. the L3 loop is probably relatively mobile. We pointed out that the presence of captopril caused an increase in the peak intensities of Trp28. Therefore, the localization spaces determined for Trp28 atoms are time averages. This was discussed in the second paragraph of Section 3.5.

Any attempt to interpret the PCSs in terms of a range of models would present a difficult inverse problem that cannot be addressed without additional assumptions. One approach, termed 'maximum allowed probability', was developed by the Florence group to identify a small number of conformations, which explain the PCS data observed for calmodulin labelled with paramagnetic lanthanide ions (Bertini et al., PNAS, 101, 6841-6846, https://doi.org/10.1073/pnas.0308641101, 2004). We are reluctant to follow this approach not only because any differences in PCSs with and without ligand are very much smaller in the case of IMP-1, but also because the PCSs were not fully reproduced in different sample preparations.

With the supply of selectively labelled indole exhausted, we are limited in the means to explore the biochemical features of IMP-1 further.

2. The minor species in the spectra seem a major concern. This problem should be explained more carefully with respect to the origin of the degradation or minor conformations and the interpretation of the data. The description in Lines 243-251 should be expanded and the interpretation should also be taken up again in the discussion, e.g. why should these minor peaks not be subconformations that correspond to some of the X-ray observations. In this respect all the additional peaks (minor species or unassigned) should be properly labeled (asterisks etc) in all the spectra, i.e. Figure 3, S3, S5, S6, S7. In this respect 'S5' on lines 245, 248 should probably be 'S6'.

Response:
As per our response to query 1, the minor peaks arise from different species, do not appear in all spectra and their cross-peaks were too weak to measure PCSs. We do not know how to prevent degradation of the protein. Unavoidably, the tagging reactions take some time and require sample handling. The legend of the new Fig. S6 states: "Stars identify weak cross-peaks arising from sample heterogeneity. They are of unknown origin and were not reproduced between different sample preparations."

For the mutant N172C with C2 tags, PCSs > 1 ppm are expected for Trp28 H$^{\varepsilon 1}$ regardless of its conformation, including the extreme conformation in the crystal structure 1DDK (green in Fig. 1). We observed no PCS of this magnitude for any of the unassigned peaks in the spectra of Fig. S5b and therefore have no evidence for alternative subconformations of the L3 loop that would be in slow exchange. We now write in the discussion (lines 423–427): "None of the minor additional cross-peaks observed in any of the sample preparations could be attributed to alternative conformations of Trp28 either. In particular, the most extreme conformation observed in the crystal structure 1DDK (green in Fig. 1) predicts PCSs > 1 ppm for Trp28 H$^{\varepsilon 1}$ in the mutant N172C with C2 tags, but we observed no PCS of this magnitude for any of the unassigned peaks."

We labelled the spectra of Fig. S5 as requested. Comparison with the selectively $^{15}$N-Trp-labelled sample of wild-type IMP-1 (now provided in Fig. S6) clearly identifies peaks originating from tryptophan. As expected, the $^{15}$N-HSQC spectra of the uniformly $^{15}$N-labelled protein are more confusing because they contain cross-peaks not only for tryptophan (Fig. S5).

The reference to Fig. S5 in lines 245-248 of ms_orig (now line 276 of ms_current) was correct. We now also refer to the new Fig. S6.

Minor:
1. It is very confusing that the number notation and the IUPAC notation are used interchangeably for the isotope labeling of the Trp sidechain and of indole. My personal preference would be IUPAC. Of course, this is your decision, but please use only one consistently. In any case, it would be very helpful to show the Trp/indole chemical structure with suitable numbering in one of the main figures.

Response:
Different IUPAC nomenclature applies to chemicals (such as indoles) and amino acids (tryptophan). To limit confusion, we added the chemical structures and their nomenclatures (as far as needed in the present manuscript) in Figure 1.

2. The labeling of all spectral figures is not very reader-friendly. It would very helpful to label the subpanels with experimental conditions such that one can immediately understand the differences. E.g. Figure 2: add mutations and experiments (NOE-relay, HSQC) to subpanel rows and columns as well as color legend. This applies to all others as well: Figure 3, S3, S4, S5, S6, S7.

Response:
We added labelling as requested.

3.line 283: the Q factors should be put into Table S7 and this Table should be referenced in the main text.

Response:
We added a column with the Q factors. Table S7 was referenced in the main text (line 273 of ms_orig and line 309 of ms_current).

4. line 302: '[13C,1H]-HSQC spectra with 150 ms NOE' seems a misnomer.

Response:
Fixed (line 336 of ms_current).

5. line 344: 'Fig. 5' is apparently 'Fig. 6'
Response:
Line 402 in ms_current: Thank you for pointing out the typo!

6. Figures S3, S4, S6: please indicate Trp sidechain assignments where available.
Response:
Figs S3, S4 and S7 in ms_current serve as overviews of the $^{15}$N-HSQC spectra. To assign all the Trp sidechain resonances in these figures would require a minute font size. Therefore why we provided the assignments of the relevant regions of the same figures in enlarged versions displayed in Figs S5 and S8. A careful comparison of the zoomed regions with the overview figures revealed an error in spectral calibration, which we amended throughout.

7. Figure S13: There is no red and blue points in the figure. They are rathe green and magenta in my PDF. Also in my impression, the localization spaces indicated by these points don't seem to be the same on left and right side.

Response:
We changed naming of the colours to magenta and cyan to be consistent with the colours in Fig. S15. (The figure is Figure S14 in ms_current.)

Thank you for considering our manuscript for publication in Magnetic Resonance.

Best regards,
Gottfried

---

## Author Response (AR3)

Dear Stephan,

In response to your comments and the comments made by Nico Tjandra, we have made the following changes.

I don't see that the lettering of Trp sidechain assignments in Figures S3, S4, S7 would have to be any smaller than for the other annotated resonances. There is plenty of space and the annotation will make the data easier to understand for readers.

Response:
We have added the assignments of the Trp sidechains in Figures S3, S4 and S7 for easier comparison with the zoomed in regions shown in Figures S5 and S8.

In response to Nico Tjandra's comments, we have made the following changes:

1. I suspect that the errors in the tensor parameters are rather small since a large number of backbone PCS were used to fit the tensor to the structure s of the protein in various forms. In fact the authors did a 20% random deletion to estimate the error. Can this standard of deviation of the parameters be added to the table S7.

Response:
The uncertainty in chemical shift measurement is rather small and, in the case of IMP-1, PCSs are more likely to be affected by variable sample conditions. We added the following footnote to Table S1: "Estimating the uncertainty of peak positions, $\sigma$, with the equation $\sigma = 0.66N/(S \times t_{2max})$, where S/N is the signal-to-noise ratio and $t_{2max}$ the acquisition time (Kontaxis et al., 2000), suggests uncertainties of the PCSs much smaller than 0.001 ppm for all PCSs measured in the present work. More critically, the chemical shifts (in particular of amide protons) are sensitive to minor differences in sample conditions between paramagnetic and diamagnetic samples, the impact of which is difficult to predict."

To obtain an estimate of minimal uncertainty ranges of the $\Delta\chi$-tensor parameters of Table S7, we produced families of tensors by random omission of 20 % of the data (as described in lines 366–368 of the main text). Table S7 is pretty congested as it is and we therefore reported the uncertainties obtained in this way in a separate table underneath (Table S8).

2. Line 436 the authors state "within the uncertainty of the experiments". So what is the uncertainty? Note that PCS as low as 1-4 ppb are listed in Table S3-S5.
I think quantifying this is quite relevant as the authors discuss the PCS analysis for W28 He1 in N172C mutant. The PCS value for that specific proton apparently is different in the two (H-C or H-N) spectra and yet using either value doesn't seem to alter the final position of that proton in the structure. This suggests to me that for that particular orientation and distance combination, large deviation in the PCS value doesn't translate to large spatial change. Yet in line 487, the authors concluded that W28 indole side chain can be determined "with remarkable accuracy". As written these sections are contradictory to me. What is the spatial accuracy?

Response:

To capture the uncertainties, we varied the $\Delta\chi$ tensors. We found the resulting localisation spaces remarkable in that they were able to distinguish clearly between different crystal structures, as pointed out in the sentences following line 487. To address the question of uncertainties, we added a new footnote to Table S1 as spelled out in our response to point 1.

Uncertainties in the $\Delta\chi$ tensors also affect the spatial accuracy, which we addressed by using a range of $\Delta\chi$ tensors fitted to variable data sets as described in lines 366 to 369. We highlight this now a bit more clearly in line 364. Furthermore, we discuss the sources of errors in a new paragraph in the discussion section as described in our response to point 3 below.

3. The authors correctly stated on line 361 that experimental error affect the PCS isosurface in a non-isotropic way. In fact, I would argue that it is really a shell with varying thickness (due to experimental error) rather than a 2D iso-surface that should be considered due to experimental error. Choosing Ln tagging site such that the area of interest is within the smallest volume of that "iso-shell" can be very important as exemplified by comment 2 above.

Response:

We thank the reviewer for the clarifying comment. We added the following paragraph to the discussion section:

"The accuracy, with which localisation spaces can be determined, further depends on the accuracy with which PCSs can be measured (which critically depends on the reproducibility of the sample conditions between the paramagnetic and diamagnetic states), the accuracy of the protein structure used to fit the $\Delta\chi$ tensors and the angle with which PCS isosurfaces of different tensors intersect. To take into account the uncertainties associated with the PCS isosurfaces, it is useful to think of each of them individually as a shell of a certain thickness (rather than a surface) that represents a compatible localisation space. Two shells of a given thickness share a smaller common space if they intersect orthogonally than if they intersect at a shallow angle."

Thank you again for considering our manuscript for publication in Magnetic Resonance.

Best regards,
Gottfried

---

## Author Response (AR4)

Dear Stephan,

Thank you for saving us from an embarrassing misuse of the uncertainty equation of Kontaxis et al. (2000). In the new version of the revised Supporting Information file, we deleted the Kontaxis reference and changed the footnote of Table S1 to a more back-of-the-envelope estimate:

"The cross-peaks of amide protons showed full linewidths at half-height of up to about 70 Hz in paramagnetic samples, while signal-to-noise ratios (S/N) typically were at least 6:1. Estimating the uncertainty of peak position as a quarter of 70 Hz yields a PCS uncertainty of 0.02 ppm. The $\Delta\chi$-tensor fits (Figure 4) suggest that actual uncertainties were of this order of magnitude or smaller. An accurate estimate of uncertainties is complicated by the sensitivity of the chemical shifts (in particular of amide protons) to minor differences in sample conditions between paramagnetic and diamagnetic samples, the impact of which is difficult to predict."

The peak positions were much better defined in the diamagnetic samples and any systematic shift between diamagnetic and paramagnetic samples were taken care of in the $\Delta\chi$-tensor fitting routine.

Best regards,
Gottfried